# TLR2 axis on peripheral blood mononuclear cells regulates inflammatory responses to non-infectious immature dengue virus particles

José Alberto Aguilar Briseño[1¤], Lennon Ramos Pereira[2], Marleen van der Laan[1], Mindaugas Pauzuolis[1], Bram M. ter Ellen[1], Vinit Upasani[1,3], Jill Moser[4], Luís Carlos de Souza Ferreira[2], Jolanda M. Smit[1], Izabela A. Rodenhuis-Zybert[1]*

1 Department of Medical Microbiology and Infection Prevention, University of Groningen and University Medical Center Groningen, Groningen, The Netherlands, 2 Vaccine Development Laboratory, Microbiology Department, Institute of Biomedical Sciences, University of São Paulo, São Paulo, Brazil, 3 Immunology Unit, Institut Pasteur du Cambodge, Institut Pasteur International Network, Phnom Penh, Cambodia, 4 Departments of Critical Care, Pathology & Medical Biology, Medical Biology section, University Medical Center Groningen, University of Groningen, Groningen, The Netherlands

¤ Current address: Department of Microbiology and Immunology, University of Iowa, Iowa City, Iowa, United States of America
* i.a.rodenhuis-zybert@umcg.nl

**Data Availability Statement:** All relevant data are within the manuscript and its Supporting information files.

## Abstract

Severe dengue virus (DENV) infection is characterized by exacerbated inflammatory responses that lead to endothelial dysfunction and plasma leakage. We have recently demonstrated that Toll-like receptor 2 (TLR2) on blood monocytes senses DENV infection leading to endothelial activation. Here, we report that non-infectious immature DENV particles, which are released in large numbers by DENV-infected cells, drive endothelial activation via the TLR2 axis. We show that fully immature DENV particles induce a rapid, within 6 hours post-infection, inflammatory response in PBMCs. Furthermore, pharmacological blocking of TLR2/TLR6/CD14 and/or NF-kB prior to exposure of PBMCs to immature DENV reduces the initial production of *inter alia* TNF-α and IL-1β by monocytes and prevents endothelial activation. However, prolonged TLR2 block induces TNF-α production and leads to exacerbated endothelial activation, indicating that TLR2-mediated responses play an important role not only in the initiation but also the resolution of inflammation. Altogether, these data indicate that the maturation status of the virus has the potential to influence the kinetics and extent of inflammatory responses during DENV infection.

## Author summary

We have previously demonstrated that Toll-like receptor 2 (TLR2), present on the surface of human monocytes, can sense DENV infection leading to the production of soluble inflammatory mediators and the activation of the endothelium. Interestingly, a large proportion of DENV particles released from the infected cells are not readily infectious

**Funding:** J.A.A.B. was supported by CONACYT, Mexico and de Cock-Hadders Foundation. I.A.R.Z. was supported by the Research Grant 2019 from the European Society of Clinical Microbiology and Infectious Diseases (ESCMID). Funding agencies had no role in the experimental design, decision to publish, or preparation of the manuscript.

**Competing interests:** The authors have declared that no competing interests exist.

because they did not complete the maturation process. Here we aimed to elucidate if and how these non-infectious, immature DENV particles contribute to systemic inflammation. We evaluated if monocytes sense immature DENV and found that sensing of immature DENV induced early inflammatory responses in PBMCs. Subsequently, by pharmacological inhibition of TLR2 and pathways downstream we demonstrated that TLR2 and CD14 drive the early production inflammatory mediators and endothelial activation. Importantly however, prolonged inhibition of TLR2 induced a second wave of TNF-α release and the subsequent activation of the endothelium. Taken together, our study showed that although virtually non-infectious, immature DENV particles can contribute to inflammatory responses and that TLR2 play an important role in their initiation and resolution. We propose that the maturation status of DENV in the human host can influence the extent and kinetics of the inflammatory responses during DENV infection.

## Introduction

Each year an estimated 390 million people contract one of the four known serotypes of the mosquito-borne dengue virus [1] (DENV1-4). Infections can be asymptomatic or lead to a self-limited febrile illness called dengue fever (DF) or dengue with warning signs [2,3]. However, in 20–25% of the symptomatic cases [4], the disease progresses to life-threatening clinical manifestations referred to as severe dengue or dengue hemorrhagic fever (DHF) and/or dengue shock syndrome (DSS) [2,3]. Severe disease is characterized by endothelial dysfunction leading to vascular permeability and plasma leakage. This is thought to be mediated by the exacerbated production of circulating inflammatory mediators such as IL-1β and TNF-α [5,6]. Despite ongoing efforts, there are currently no methods available to predict and/or prevent disease progression, which demonstrates that elucidation of the mechanism(s) regulating the excessive inflammatory responses elicited by DENV is crucial for the understanding of its pathogenesis and the proper handling of the severe cases.

The innate immune system represents the first line of defense against invading pathogens [7]. By expressing a variety of pattern recognition receptors (PRR) such as toll-like receptors (TLR), innate immune cells can sense and respond to pathogen-associated molecular patterns (PAMP's) and tissue-derived danger-associated molecular patterns (DAMP's). Such recognition initiates a cascade of intracellular pathways leading to the release of endothelium activating cytokines and chemokines [8,9], which ultimately aid in containing the infection. However, uncontrolled production of inflammatory mediators, and, in turn, exacerbated activation of endothelial cells can result in disruption of their integrity and plasma leakage, the hallmarks of severe DENV infection [10,11]. To date, several PRRs have been implicated in sensing of DENV infection. For instance, endosomal TLR3 and TLR7, and cytosolic retinoic acid-inducible gene I (RIG-I) and melanoma differentiation-associated protein 5 (MDA5) have been shown to sense the DENV RNA early in the infection, thereby leading to the production of type I interferons [12–14]. In contrast, plasma membrane-expressed TLR4-mediated sensing of the soluble form of DENV non-structural protein 1 (NS1) has been reported to contribute to the loss of endothelial integrity [15,16]. Additionally, our recent study uncovered that cell-surface expressed TLR2, and its co-receptors CD14 and TLR6, act as an early sensor of DENV infection in primary blood mononuclear cells *in vitro*. Furthermore, in acute dengue patients increased TLR2 expression on monocytes is predictive of severe disease development [17].

DENV is an enveloped virus with a single-stranded RNA genome of positive polarity belonging to the family *Flaviviridae*. The envelope of mature DENV virions anchors two glyco-proteins, envelope (E) and membrane (M) [18]. The E glycoprotein mediates cell entry and the low pH-dependent fusion of the viral and endosomal membrane. Upon this fusion, the viral genome is released into the cytoplasm [19,20]. Following protein translation and RNA replica-tion, the virus progeny containing the E and precursor M (prM) glycoproteins, are assembled in the ER. Presence of prM protein on newly assembled particles protects E from premature fusion from within the acidic compartments of exocytotic pathway. The immature virions mature while passing through the trans-Golgi network where the host enzyme furin cleaves prM into M and the pr peptide [18,21,22]. The pr peptide is retained on the mature virus prog-eny throughout exocytosis and dissociates from the virus particle after its release to the extra-cellular milieu [21,23–25]. Interestingly, in the case of DENV, cleavage of prM into M is often inefficient and a mixture of heterogeneous particles encompassing mature, partially mature and fully mature forms [26] are secreted. Depending on the cell-type, up to 40% of released DENV virions still contain prM [22,27].

In general, fully immature virions are considered to be low or non-infectious [27–29]. Importantly, however, despite their residual infectivity, the circulating immature virions are likely to be sensed by immune cells [29,30] and thereby, at least in part, contribute to innate immune responses triggered upon infection. Indeed, Décembre and colleagues showed that plasmacytoid dendritic cells can sense immature DENV particles leading to a potent produc-tion of IFN-$\alpha$ [30]. Notably, the contribution of monocytes and macrophages in the sensing of immature virions and subsequent inflammatory responses is yet to be elucidated. In this study, we characterized the early inflammatory response triggered by immature DENV particles and scrutinized the role of TLR2 in these responses using *ex vivo* systems based on primary human PBMCs and endothelial cells. Our data uncover the ability of immune cells to readily sense immature DENV virions in a TLR2 axis-dependent manner.

## Results

### Immature DENV2 particles induce a rapid onset of inflammatory responses

DENV particles are secreted from infected cells as a mixture of fully mature, partially imma-ture and fully immature particles. We have previously characterized fully immature DENV particles as low or non-infectious [27–29]. However, their role in inflammation during DENV infection remain elusive. To test the ability of immature DENV virions to induce inflammatory responses in PBMCs that contribute to the vascular activation, we have used the previously established model in endothelial cells (EC) [31]. Briefly, primary human umbilical vein endo-thelial cells (HUVEC) were incubated with supernatants from PBMCs (three donors) exposed to fully immature virus preparations of DENV2 (prM-DENV2) at a multiplicity of genomes (MOG) of 300, which for infectious DENV corresponds to a multiplicity of infection ~5, for 6h, 12h, 24h, 48h, and 72h. Supernatants from PBMCs infected at a MOG 300 with standard (std) DENV2 were used as a comparison. Supernatants harvested from mock-treated PBMCs at the corresponding time points served as a reference control. After 6 hours of stimuli, the surface expression of E-selectin, vascular cell adhesion protein 1 (VCAM-1) and intercellular adhesion molecule 1 (ICAM-1) was determined by flow cytometry and analyzed according to the gating strategy depicted in S1A Fig. To ensure that any measured responses are due to the sensing of virions by PBMCs rather than the direct effect of the virus particles on the EC, HUVEC were treated with an equal number of MOG as those present in the supernatants of prM-DENV2 treated PBMCs. HUVEC incubated with LPS served as a positive control. The

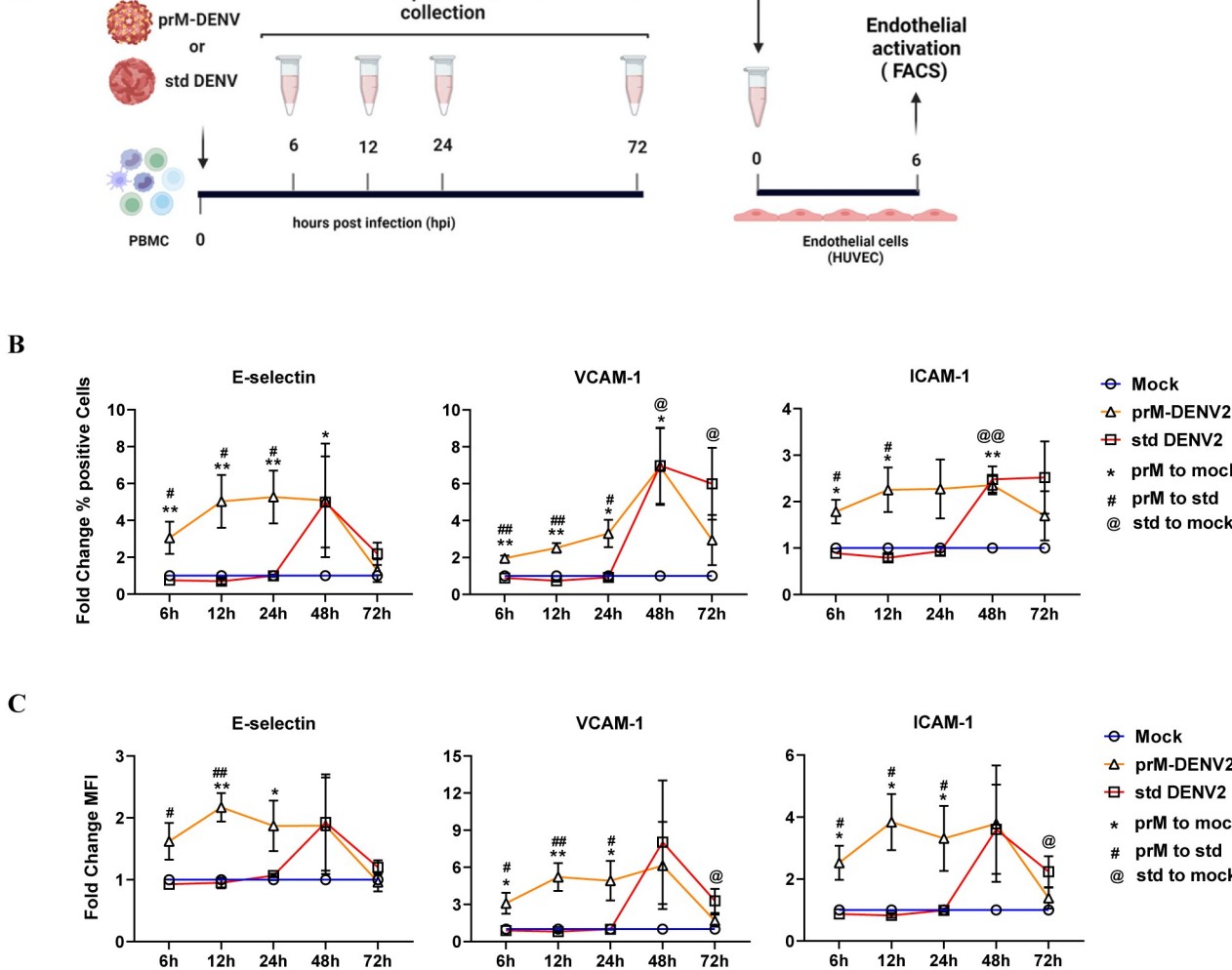

**Fig 1. prM-DENV2 induces an early onset of soluble inflammatory responses.** (**A**) Experimental scheme: HUVEC were incubated with cell-free supernatants harvested at indicated time points from hPBMCs (n = 3, three different donors) exposed to prM-DENV2 (MOG 300), std DENV2 (MOG300) or mock treatment. Surface expression of E-selectin, VCAM-1 and ICAM-1 on HUVEC was determined by flow cytometry and represented as (**B**) percentage of positive cells and (**C**) MFI normalized to relative mock values. Data represents the mean ± SEM. P values were obtained by unpaired one-tailed t test ($^{*/\#/@}$P<0.05; $^{**/\#\#/@@}$P<0.01). (**A**) Figure created with BioRender.com.

results show that LPS induced endothelial activation whereas immature virions did not (S1B Fig). Notably, the supernatants of prM-DENV2-challenged PBMCs induced the activation of EC as evidenced by a significant increase in the percentage of positive cells (Fig 1B and S2A Fig) and MFI (Fig 1C and S2B Fig) of adhesion molecules compared with their respective mock-treated and std DENV2-infected conditions. Furthermore, the soluble inflammatory responses induced by prM-DENV2 were transient with a peak response between 6–24 hpi and resolution was seen between 48–72 hpi (Fig 1B and 1C). In contrast, and as previously observed [17], std DENV2 induced an onset of soluble inflammatory responses that peaked at 48h and start resolving at 72h (Fig 1B and 1C). Importantly, and in line with our previous study [28], prM-DENV2 was virtually non-infectious in PBMCs, as evidenced by the absence of DENV-E positive cells and lack of infectious virus in PBMCs' supernatants even after 72 hours of infection, which in theory corresponds to 3 replication cycles (S3A Fig including std DENV2 and UV-prM-DENV2 as a positive and negative controls, respectively). In agreement

with our surrogate assay, std DENV2 but not UV- prM-DENV2, UV-std DENV2 or prM-DENV2 was infectious on HUVEC as evidenced for the presence of viral antigens (E and NS3, S3B and S3C Fig). This implies that the observed kinetics of innate responses induced by prM-DENV2 are not modulated by virus replication. Altogether, our data show that prM-DENV2 can induce a rapid onset of inflammatory mediators in PBMCs, which in turn can activate endothelial cells.

## Immature DENV virus particles trigger NF-κB activation in a TLR2/6 and CD14 dependent manner

We have recently shown that DENV infection-induced immune responses *in vitro* are controlled by TLR2/6 and CD14 [17]. In that study, we used DENV preparations produced in human or mosquito cells referred to herein as standard (std). These preparations, especially the latter, contain a substantial amount of virions with different degrees of maturation [22,26,27,32,33]. Hence, we hypothesized that the TLR2 axis could also sense immature DENV virus particles. To test this, we have first used the HEK-Blue hTLR2 reporter cells co-expressing TLR2, its co-receptors TLR1/6/CD14 and the SEAP (NF-kB/AP1- inducible secreted embryonic alkaline phosphatase) genes. Consequently, TLR2-mediated activation of NF-κB and AP1 can be quantified by the colorimetric analysis of SEAP in the cell supernatant. In all experiments, stimulation of the cells with TLR2/TLR1/CD14 agonist PAM3CSK4 (PAM3) served as a positive control. As shown in Fig 2A, immature DENV2 potently activate HEK-Blue hTLR2 cells in a concentration-dependent manner, whereas no activation was observed in the parental HEK-Blue Null1 cells (S4A Fig). Purified prM-DENV2 activated NF-κB to the same extent demonstrating that molecular patterns present on the surface of the virus rather than a soluble factor are responsible for this activation (S4B Fig). Importantly, also immature preparations of DENV1 and DENV4 particles activated NF-κB (S4C Fig). The NF-κB activation induced by immature DENV2 was significantly reduced upon the block of TLR2, TLR6, and CD14 but not that of TLR1(Fig 2B). Unfortunately, due to the low titers of the prM-DENV1 and prM-DENV3 preparations growth in furin-deficient LoVo cells, we could not test MOG250 (prM-DENV3) and MOG1000 (prM-DENV1 and prM-DENV3), as tested for prM-DENV2 and prM-DENV4. These data suggest that prM-containing DENV virions can be sensed in a TLR2/TLR6/CD14 dependent manner.

To assess if TLR2 axis-mediated sensing of immature virions occurs at the plasma membrane or in an endosomal compartment, we interrogated its dependency on clathrin-mediated endocytosis. To this end, we tested the effects of pit-stop (PS), an inhibitor of clathrin-mediated endocytosis [34]; $NH_4Cl$, a drug that neutralizes the pH of intracellular compartments [35] or wortmannin (WN), an inhibitor of macropinocytosis and phagocytosis [36], on TLR2-mediated NF-κB activation induced by immature virions. PAM3, which has been shown to require clathrin-mediated endocytosis to trigger NF-κB activation [37] served as a control. Interestingly, similarly to what was observed for PAM3, PS and $NH_4Cl$ significantly decreased the activation of NF-κB induced by prM-DENV2 while wortmannin had no effect on its activation (Fig 2C). These data are in line with previous findings [17,37–40] and support the notion that internalization of the TLR2/TLR6/CD14 complex via clathrin-mediated endocytosis potentiates subsequent activation of NF-κB.

## Production of inflammatory mediators in PBMCs in response to immature virions relies on TLR2/TLR6/CD14 mediated NF-κB activation

We next sought to verify if the TLR2/TLR6/CD14 axis is responsible for the early secretion of soluble inflammatory mediators by PBMCs in response to prM-DENV2 (Fig 3). We analyzed

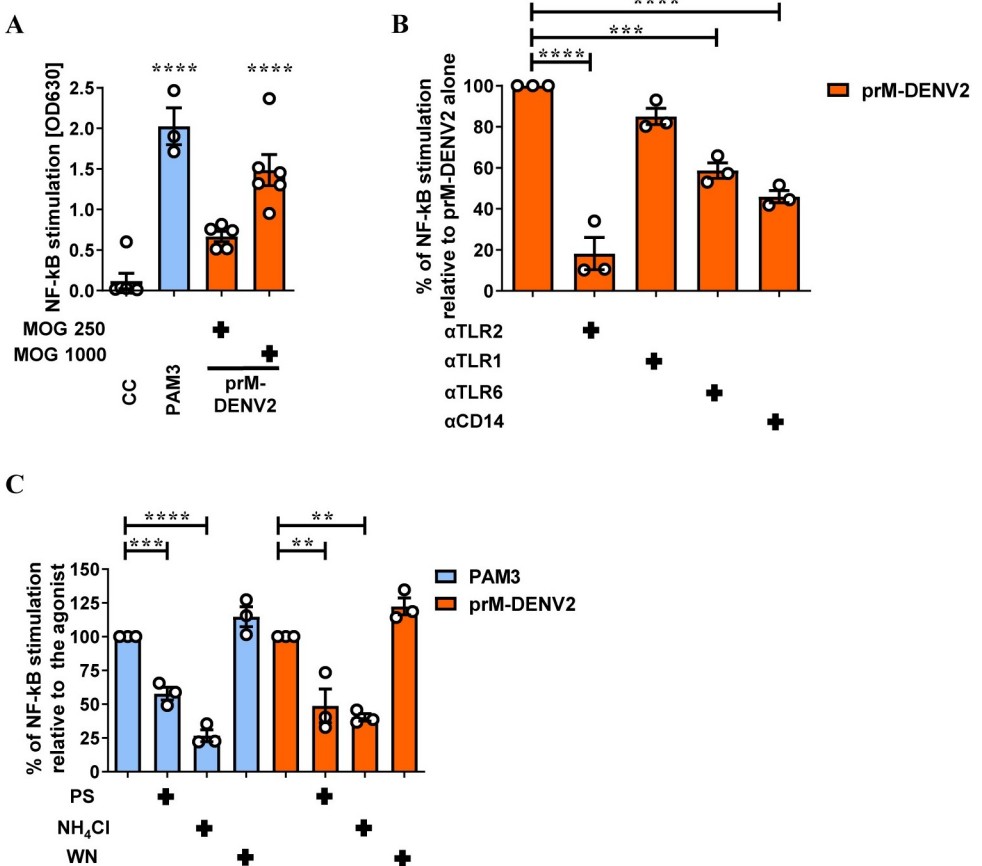

**Fig 2. Immature DENV2 virions engage TLR2/6/CD14 to activate NF-κB.** (**A**) NF-κB activation in HEK-Blue hTLR2 cells (mock)-treated with PAM3CSK4 (PAM3, 50 ng/mL, n = 3) and prM-DENV2 (MOG 250, n = 5; MOG 1000, n = 6) for 24h. (**B**) NF-κB activation in HEK-Blue hTLR2 cells pretreated for 2h with αTLR2, αTLR1, αTLR6 and αCD14 (15 μg/mL) before exposure to prM-DENV2 (MOG1000) for 24h, n = 3. (**C**) NF-κB activation in HEK-Blue hTLR2 cells pretreated for 1h with endocytosis inhibitors pitstop (PS, 60μM), ammonium chloride (NH$_4$Cl, 50mM) and wortmannin (WN, 2 μM), before exposure to prM-DENV2 (MOG1000) for 24h, n = 3. NF-κB stimulation was assessed by QUANTI-Blue, OD values show the induction of NF-κB. Data represents the mean ± SEM. P values were obtained by one-way ANOVA, Dunnett post hoc test (**P<0.01; ***P<0.001; ****P<0.0001). n = independent biological experiments. CC: cellular control. PAM3 and prM-DENV2 as a control of their respective blocking/ treatment conditions.

a total of 13 cytokines including IL-8, IL-10, IL-12p70, IP-10, GM-CSF, IFN-α2, IFN-β, IFN-γ, IFN-λ1, and IFN-λ2/3 in the supernatants of PBMCs exposed to prM-DENV2 in the presence or absence of TLR2 axis blocking antibodies. Interestingly, sensing of prM-DENV2 triggered the production of TNF-α, IL-1β, and IL-6 cytokines which are usually associated with EC activation and damage during severe dengue (Fig 4A and 4B) [11,41]. Furthermore, blocking TLR2/6 and CD14 reduced the production of these cytokines (Fig 4A and 4B) while TLR1 block (or the use of αTLR2/ αTLR1/6 isotype controls) did not affect these responses (S5 Fig). Notably, in contrast to the recently reported TLR2-mediated cytokinome induced by std DENV2 [17], prM-DENV2 did not induce considerable levels of antiviral IFNs (IFN-α2, IFN-β, IFN-λ1) (S6 Fig). PrM-DENV also did not induce substantial changes in the levels of IL-12p70, IP-10, GM-CSF, IFN-γ and IFN-λ2/3. Finally, the TLR2 axis block did not affect the levels of IL-8, IL-10, IFN-β and IFN-λ1 (S6 Fig).

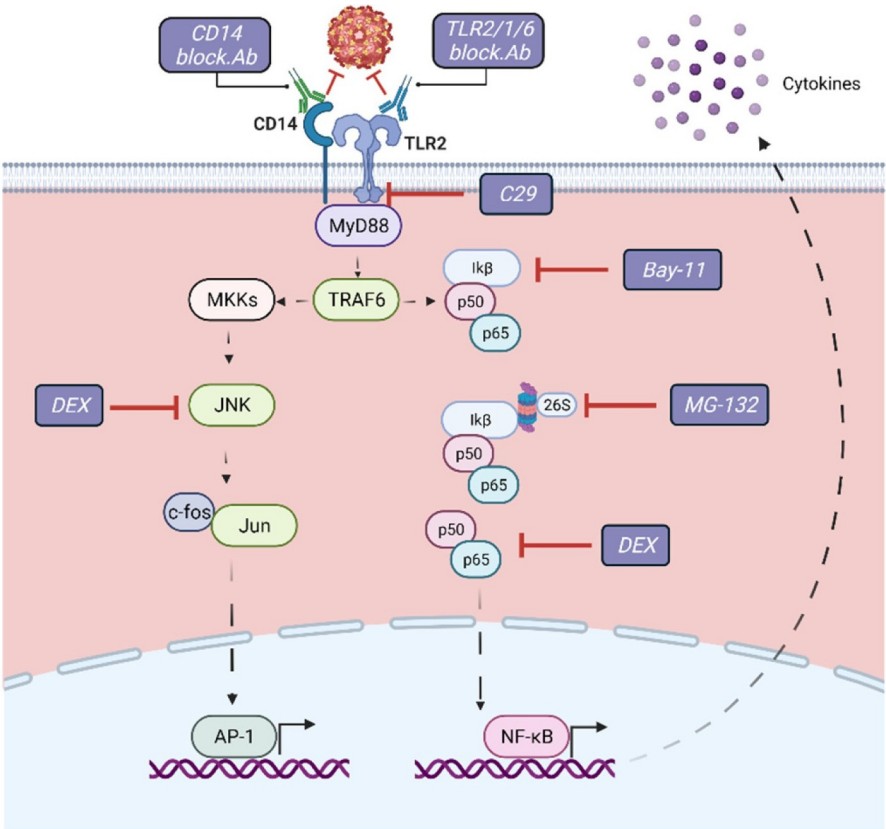

**Fig 3. Schematic representation of the signaling pathways and molecules subjected to pharmacological targeting downstream TLR2 axis.** C29 (C16H15NO4) is a TLR2 inhibitor that blocks the interaction of the receptor TIR domain with MyD88; Dexamethasone (DEX) blocks NF-κB and MAPK activation during TLR engagement; Bay-11inhibits the phosphorylation of IkB which results in the inactivation of NF-kB; MG-132 inhibits the degradation of IκB by blocking the proteolytic activity of the 26S proteasome. Figure was created with BioRender.com.

To attest the role of TLR2 in the mediated production of cytokines and, at the same time, identify the cell type responsible for cytokine production, we measured the intracellular accumulation of IL-1β and TNF-α in PBMCs exposed to prM-DENV2 in the presence or absence of TLR2 axis block. To do this, PBMCs were harvested 12hpi, the time-point corresponding to the peak of EC activation (Fig 1B and 1C) and analyzed by flow cytometry. The intracellular accumulation of both cytokines was measured in the live monocyte and lymphocyte fractions of the PBMCs gated based on size and granularity, following the gating strategy shown in S7A Fig. The intracellular accumulation of IL-1β and TNF-α following prM-DENV2 was found primarily in monocytes rather than lymphocytes, when compared to the mock-treated conditions (Fig 4C and S7B Fig). Moreover, and in line with the LegendPlex data, blocking TLR2/TLR6 but not TLR1 or the use of isotype control significantly reduced the intracellular accumulation of TNF-α and IL-1β, while the CD14 block moderately decreased the accumulation of both cytokines (Fig 4C and S7C Fig). Altogether, these results indicate that TLR2/TLR6/CD14- mediated sensing of immature dengue particles induces the production of TNF-α and IL-1β by monocytes.

To gain more insight into the mechanism underlying the TLR2-mediated production of TNF-α and IL-1β in monocytes, we pharmacologically targeted key molecules of signaling downstream of TLR2 prior to infection (Fig 3). We first analyzed the influence TLR2 axis on

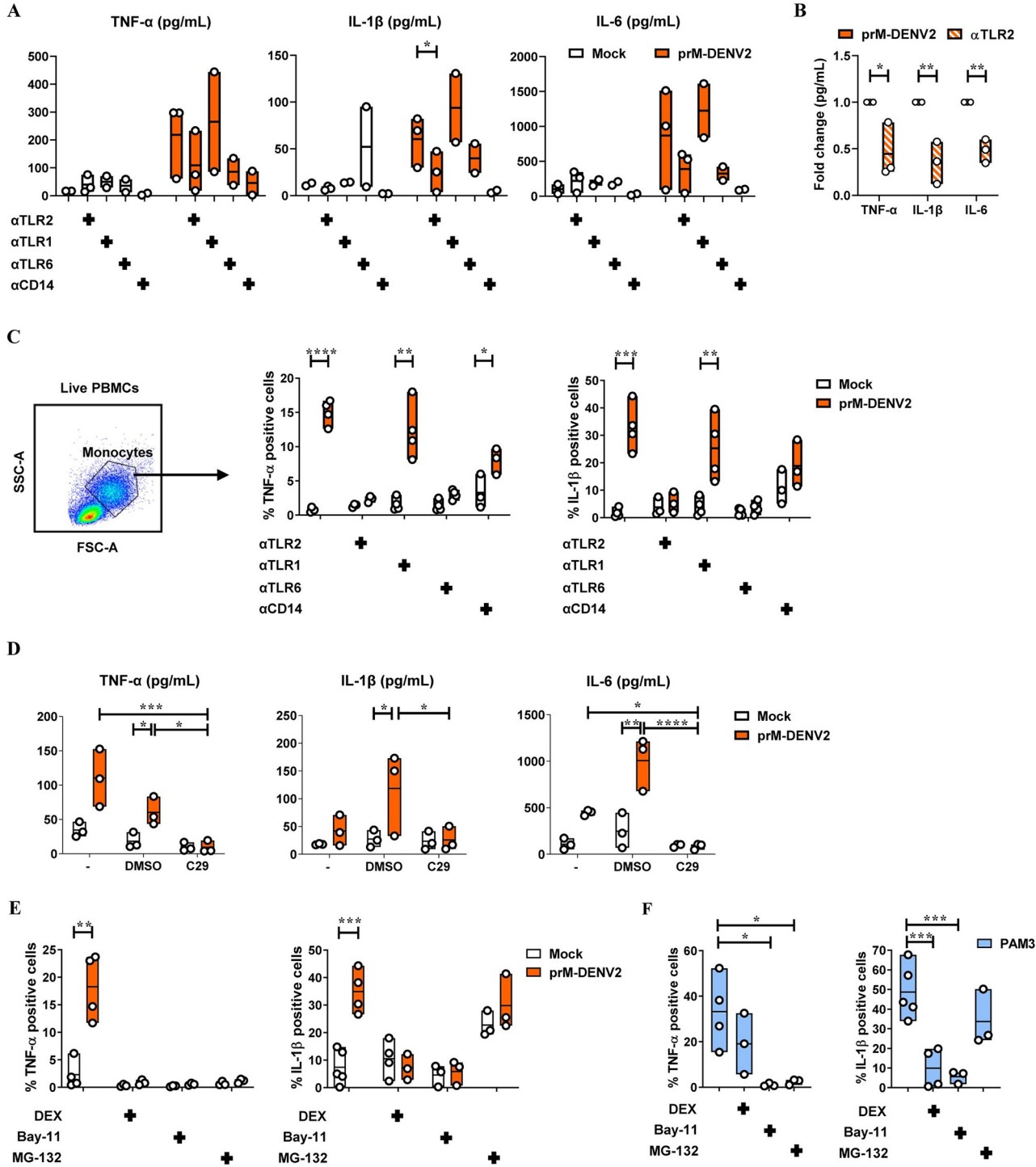

**Fig 4. TNF-α- and IL-1β intracellular accumulation induced by prM-DENV2 is TLR2/6/CD14 and NF-κB mediated.** (**A, B** and **D**) PBMCs from healthy donors (n = 3–4, three to four different donors) were (mock) treated with αTLR2 (5 μg/mL), αTLR1 (5 μg/mL), TLR6 (5 μg/mL), αCD14 (3 μg/mL), C29 (100 μM) or its solvent control (DMSO) for 2 hours prior exposure with prM-DENV2 at MOG 300 for 6h. Cytokine production (in picograms per milliliter (pg/mL)) in the PBMCs' supernatants was measured by flow cytometry using LegendPlex. (**C, E** and **F**) PBMCs from healthy donors (n = 3–5, three to five different donors) were (mock) treated with αTLR2 (5 μg/mL), αTLR1 (5 μg/mL), TLR6 (5 μg/mL), αCD14 (3 μg/mL), DEX, (10μM), Bay-11 (5μM) and MG-132 (9.5 μg/mL) for 2 hours prior exposure with prM-DENV2 at MOG 300 or PAM3CSK4 (PAM3, 600 ng/mL) for 12h in the presence of Brefeldin-A. (**C, E** and **F**) Percentage of monocytes (in PBMCs) with intracellular expression of TNF-α and IL-1β was measured by flow cytometry. P values were obtained by unpaired one-tailed t test. (*P<0.05, **P<0.01, ***P<0.001, ****P<0.0001).

the prM-DENV2 sensing using the C29 compound ($C_{16}H_{15}NO_4$), a TLR2 inhibitor that blocks interaction of the receptor TIR domain with MyD88[42] (Fig 3). Accordingly, PBMCs were treated with C29, followed by exposure to prM-DENV2 for up to either 6h, 24h or 48 hours. Since this compound appeared moderately toxic to cells when kept in culture for longer than 24h (S8A Fig), we evaluated the cytokine production of PBMCs exposed to prM-DENV2 at early time-point (6hpi). Interesting, the production of TNF-α, IL-1β and IL-6 cytokines from prM-DENV2-stimulated PBMCs was significantly blocked by C29 treatment (Fig 4D). In addition, the EC activation measured by surface expression of adhesion molecules (E-selectin, VCAM-1 and ICAM-1) on HUVECs treated with supernatants from C29-treated PBMCs exposed to prM-DENV2 or PAM3 was also reduced (S8B, S8C and S9 Figs). Thus, these results corroborate the findings that the TLR2 axis mediates the detection of prM-DENV2 in PBMCs and the consequent induced inflammatory response.

Effectively, we blocked NF-κB and/or AP-1, two central regulators of the immune response downstream of TLR signaling [43–45]. Specifically, we used dexamethasone (DEX), a drug targeting pathways leading to the activation of both NF-κB and AP-1 [46]; Bay-11-7080 (Bay-11), an inhibitor of the IκB-α phosphorylation upstream of NF-κB [47] and MG-132, a selective inhibitor of the proteolytic activity of the 26S proteasome involved in the degradation of IκB, upstream of NF-κB [48] (Fig 3). The TLR2 agonist PAM3 served as a reference to compare the pathways involved in the production of cytokines downstream of the TLR2 axis. Briefly, PBMCs were treated with the indicated inhibitors for 2h prior to their exposure to prM-DENV2 or PAM3 for 12h in the presence of brefeldin-A. The presence of the inhibitors and blocking antibodies, did not affect PBMCs viability (S7D and S7E Fig). The intracellular accumulation of IL-1β and TNF-α was then determined by flow cytometry following the gating strategy shown in S7A Fig. Furthermore, as expected, prM-DENV2 did not induce the accumulation of the cytokines in the lymphocyte fraction regardless of the presence of the inhibitors (S7F Fig). We thus focused our analysis on monocytes (gated based on size and granularity) (Fig 4C). Interestingly, DEX significantly reduced the accumulation of both cytokines induced by prM-DENV2 (Fig 4E). However, in response to PAM3, DEX inhibited the accumulation of only IL-1β while no effect on TNF-α was observed (Fig 4F). This highlights the diversity and agonist-specificity of signaling pathways downstream of TLR2 [45]. Bay-11, which selectively targets the NF-κB signaling pathway, significantly abrogated the accumulation of both cytokines induced by prM-DENV2 and PAM3 (Fig 4E and 4F). MG-132 abrogated the accumulation of TNF-α and IL-1β induced by prM-DENV2 and PAM3. Notably, MG-132 moderately induced the accumulation of IL-1β in non-stimulated cells (Fig 4E), suggesting that the proteasome regulates the basal levels of production of IL-1β [49,50]. Altogether, the data indicate that the TLR2/TLR6/CD14-mediated sensing of prM-DENV2 leads to the activation of NF-κB and the subsequent production of TNF-α and IL-1β by monocytes.

## TLR2 axis initiates early and dampens the late soluble inflammatory responses to immature DENV2

Since TLR2-mediated sensing of prM-DENV2 led to production of at least two cytokines commonly implicated in EC activation during inflammation [51], we tested if TLR2 block would indeed abrogate endothelial activation induced upon sensing of prM-DENV2 in PBMCs. Accordingly, we analyzed changes in the surface expression of E-selectin, VCAM-1 and ICAM-1 on HUVEC following 6h treatment with cell-free supernatants from PBMCs exposed to prM-DENV2 in the presence or absence of TLR2 axis blocking antibodies or TLR2 isotype control (Fig 5A). Since the soluble inflammatory responses induced by prM-DENV2 peaked

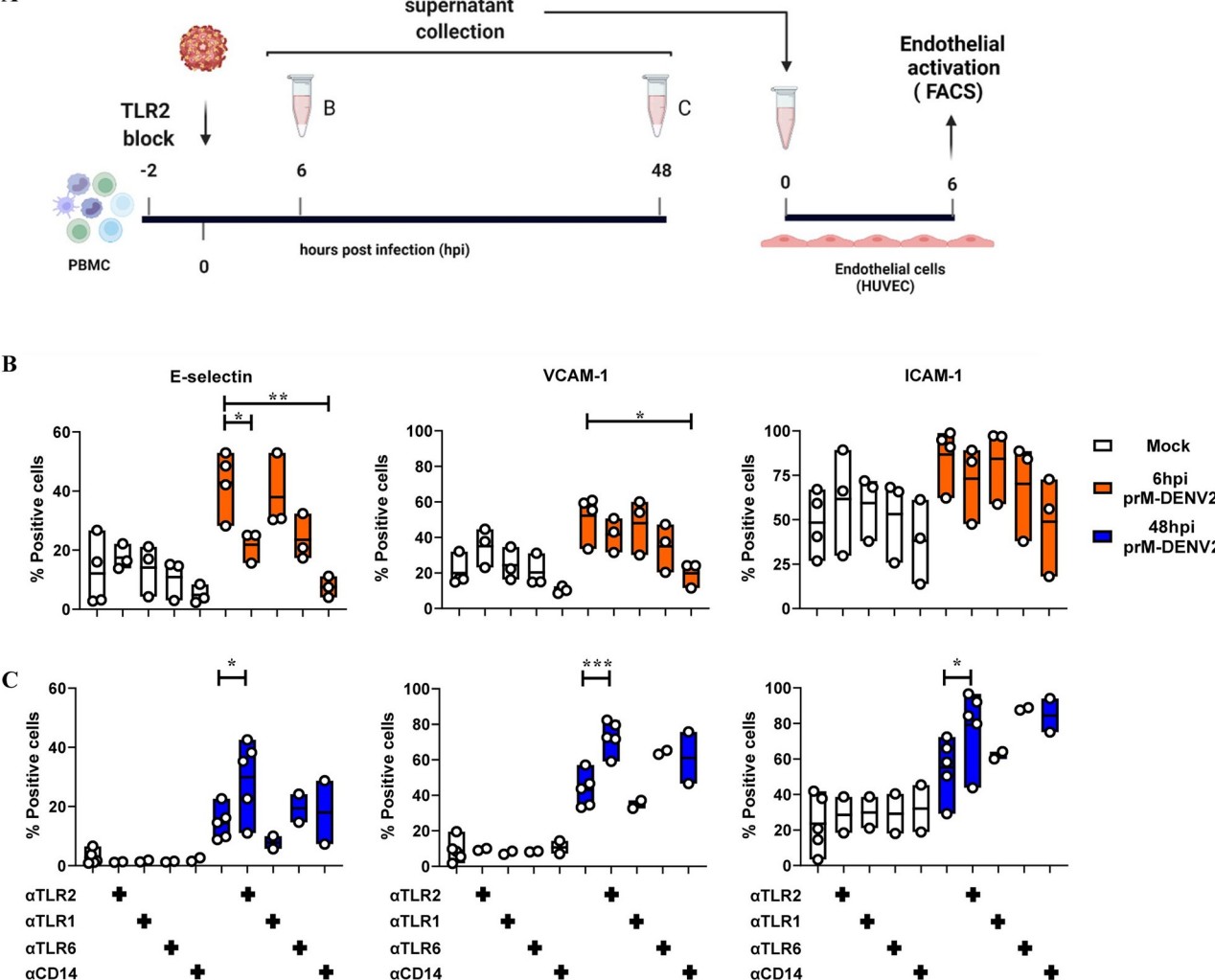

**Fig 5. TLR2/TLR6/CD14 axis controls the early, but not the late onset of soluble inflammatory responses induced by prM-DENV2.** (**A**) HUVEC were incubated for 6h with cell-free supernatants from PBMCs (n = 2–5, two to five different donors) (mock)-exposed for (**B**) 6h and (**C**) 48h to prM-DENV2 at a MOG of 300 in the presence or absence of TLR2/TLR1/TLR6/CD14 blocking antibodies. Surface expression of E-selectin, VCAM-1 and ICAM-1was determined by flow cytometry and represented as percentage of positive cells. P values were determined by unpaired one-tailed t-test. (*P<0.05, **P<0.01, ***P<0.001). (**A**) Figure created with BioRender.com.

between 6–24 hpi and were solved between 48–72 hpi (Fig 1B and 1C), we chose to evaluate the effect of the TLR2/TLR6/CD14 axis block at 6hpi (Fig 5B) and 48hpi (Fig 5C). Interestingly, the supernatants from prM-DENV2-treated PBMCs in the presence of TLR2 or CD14 blocking antibodies significantly reduced the percentage of E-selectin positive EC, while a marginal decrease was observed when TLR6 was blocked (Fig 5B). Moreover, a marginal decrease in the cell frequencies of VCAM-1 and ICAM-1 was observed when EC were exposed to supernatants from prM-DENV2 treated PBMCs in the presence of TLR2/TLR6/ CD14 but not TLR1 blocking antibodies or isotype control (Fig 5B and S10A Fig). The low levels of TNF-α and IL-1β found in the supernatants from prM-DENV2-treated PBMCs in the presence of αTLR2 and αCD14 antibodies (Fig 4A) may explain the significant reduction of E-selectin positive EC (Fig 5B), as both cytokines are commonly associated with the upregulation of this adhesion molecule [51]. Of note, the supernatants from mock infected PBMCs treated

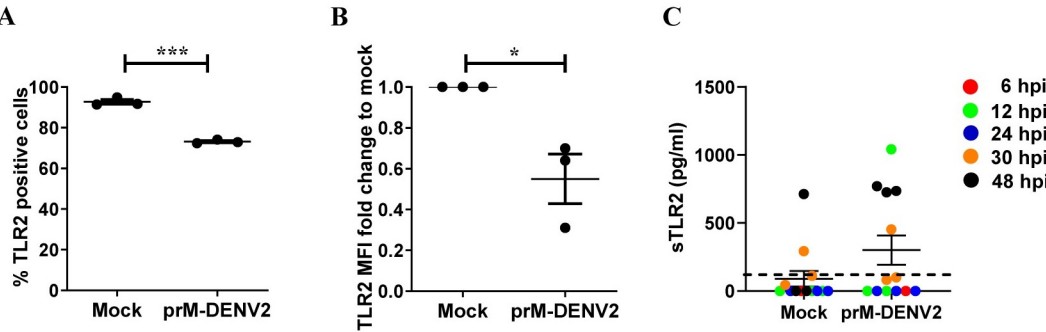

**Fig 6. prM-DENV2 downregulates TLR surface expression on monocytes within the PBMCs.** PBMCs from healthy donors (n = 3, three different donors) were exposed to prM-DENV2 at MOG 300 for 6h, 12h, 24h, 30h and 48h. (**A**) Percentages of TLR2 positive monocytes were determined by flow cytometry. (**B**) Fold changes in surface expression of TLR2 relative to the mock. (**C**) Concentration of sTLR2 in the supernatants exposed to prM-DENV2 determined by ELISA. Error bars represent mean± SEM. P values were obtained by one-tailed t test. (*P<0.05; ***P<0.001).

with TLR2/TLR1/TLR6/CD14 blocking antibodies or isotype control had no effect on the expression of the adhesion molecules (% positive cells) in EC (Fig 5B and S10 Fig). Additionally, a similar effect of TLR2 blocking was observed when EC were treated with supernatants from PBMCs exposed to prM-DENV1 and prM-DENV4 (S11A and S11B Fig).

Surprisingly, however, while TLR2 block mitigated the inflammatory responses of PBMCs at 6hpi, the opposite effect was observed at 48hpi time-point. There, the presence of TLR2 block during prM-DENV2 exposure significantly increased the frequencies of the three adhesion molecules on EC (Fig 5C), similar responses were observed when blocking TLR6 and/or CD14. This was specific to TLR2/TLR6/ CD14 axis as blocking of TLR1 or isotype control showed no such effect (S10B Fig). Interestingly, a similar increase in the late inflammatory responses was observed for prM-DENV1 and prM-DENV4 in the presence of the TLR2 block, while the blocking of CD14 minimally decreased the responses elicited by prM-DENV4 (S12A and S12B Fig). Altogether, these results suggest that TLR2 functions not only in the initiation of inflammation but also its resolution. Since the reduction of TLR2 expression in the course of inflammation through e.g. shedding of the decoy, soluble sTLR2, may dampen inflammation[52], we measured the surface expression of TLR2 on monocytes (within the PBMCs) after 48 hpi with prM-DENV2. The expression of TLR2 was significantly decreased after 48h exposure with prM-DENV2 as evidenced by the decrease in the percentage of TLR2 positive monocytes and MFI when compared to the mock-treated cells (Fig 6A and 6B). To determine whether the lower expression of TLR2 on monocytes correlates with shedding, we measured the concentrations of soluble TLR2 (sTLR2) in supernatants from PBMCs treated with prM-DENV2 for 6–48 hours. Indeed, the release of sTLR2 was detected predominantly in the supernatants from prM-exposed PBMCs harvested at 48hpi while low to baseline (mock) levels of sTLR2 were found in those from 6 to 24hpi (Fig 6C). This finding suggests that downregulation of the surface levels of TLR2 inter alia via shedding could contribute to controlling the extent and kinetics of inflammation during DENV infection. Further studies are needed to validate this premise.

## TNF-α released from prM-DENV2-exposed PBMCs is a soluble mediator that drives endothelial cell activation

Finally, we sought to better understand the mechanism underlying the increase of EC activation observed after prolonged TLR2 blockage. To this end, we examined the kinetics of

intracellular accumulation of TNF-α and IL-1β in the prM-DENV2-treated PBMCs from 6 to 48 hpi (Fig 7A). In accordance with the data presented in Fig 4C, an increased frequency of TNF-α and IL-1β positive cells was observed at 6 hpi in the prM-DENV2-treated monocytes (within PBMCs) and was drastically reduced upon TLR2 blockage (Fig 7B). Interestingly however, at late time-point (24–48 hpi), the TLR2 blockage resulted in a significant increase of TNF-α -positive monocytes while, IL-1β accumulation remained low throughout duration of the experiment (Fig 7A and 7B). Furthermore, no significant changes were observed for cytokine accumulation in the lymphocyte population, independent of the time-point or blocking conditions (S13 Fig). Altogether, these results suggest that TNF-α produced by monocytes at late time-points may be related to the increased EC activation profile. To test this hypothesis, we neutralized the activity of TNF-α in the supernatants prior to their incubation with HUVECs using anti-TNF-α antibody[53] (Fig 7C). The potency of the utilized protocol is shown in S14 Fig. Depletion of TNF-α in the supernatants harvested 6h after exposure of PBMCs to prM-DENV2 resulted in a marginal reduction of ICAM-1 and VCAM-1 expression on HUVEC (Fig 7D), suggesting that more cytokines contribute to the vascular responses observed at this time point. However, when TNF-α was depleted from the supernatants harvested at 48h following exposure to prM-DENV2 under TLR2 block conditions, the increased EC activation was completely abolished (Fig 7E). Thus, prolonged TLR2 block disturbed the process of inflammation resolution by ultimately leading to the induction of TNF-α and increased activation of ECs. Altogether, these data underscore the pivotal role of monocyte expressed TLR2 in the sensing of immature dengue particles and subsequent modulation of inflammatory responses (Fig 8).

## Discussion

In this study, we have characterized the kinetics and mechanism of the inflammatory responses elicited upon sensing of immature DENV particles. By employing human primary *ex vivo* systems, namely PBMCs and HUVEC, we show that the sensing of immature virions by monocytes induces an early onset of soluble inflammatory responses leading to the activation of ECs. The production of several inflammatory mediators including IL-1β and TNF-α was found to be under control of TLR2 and its co-receptors TLR6 and CD14 and relied on the activation of the transcription factor NF-κB. Interestingly, however, prolonged exposure of PBMCs to immature particles in the presence of TLR2/TLR6 block augmented production of TNF-α resulting in increased activation of ECs. Altogether, our data provide evidence for the ability of TLR2 on monocytes to not only sense immature dengue particles but also regulate the subsequent inflammatory responses.

In the current study, we focused on the contribution of fully immature dengue particles to the innate inflammatory responses in course of the infection. Importantly, *in vitro*-produced DENV preparations are known to contain a mixture of particles with different degree of maturation encompassing immature, partially mature and immature virions [22,26,27,32]. Thus, the maturation state of DENV particles might impact the overall immune responses during infection. Interestingly, a recent study by Raut *et al.*, showed that more infectious and mature DENV1 particles prevail in blood samples of infected patients when compared to that produced in cell culture [32]. Conversely, the presence of prM antibodies has been observed generally in patients with secondary infection rather than those with primary infection [54], suggesting that immature DENV particles do circulate in patients. Interestingly, partially mature DENV particles display both mature and immature E protein conformations and are infectious [55]. Thus, further studies should compare the differential innate immune responses induced by *in vivo*-circulating DENV serotypes to those elicited by *in vitro* produced DENV.

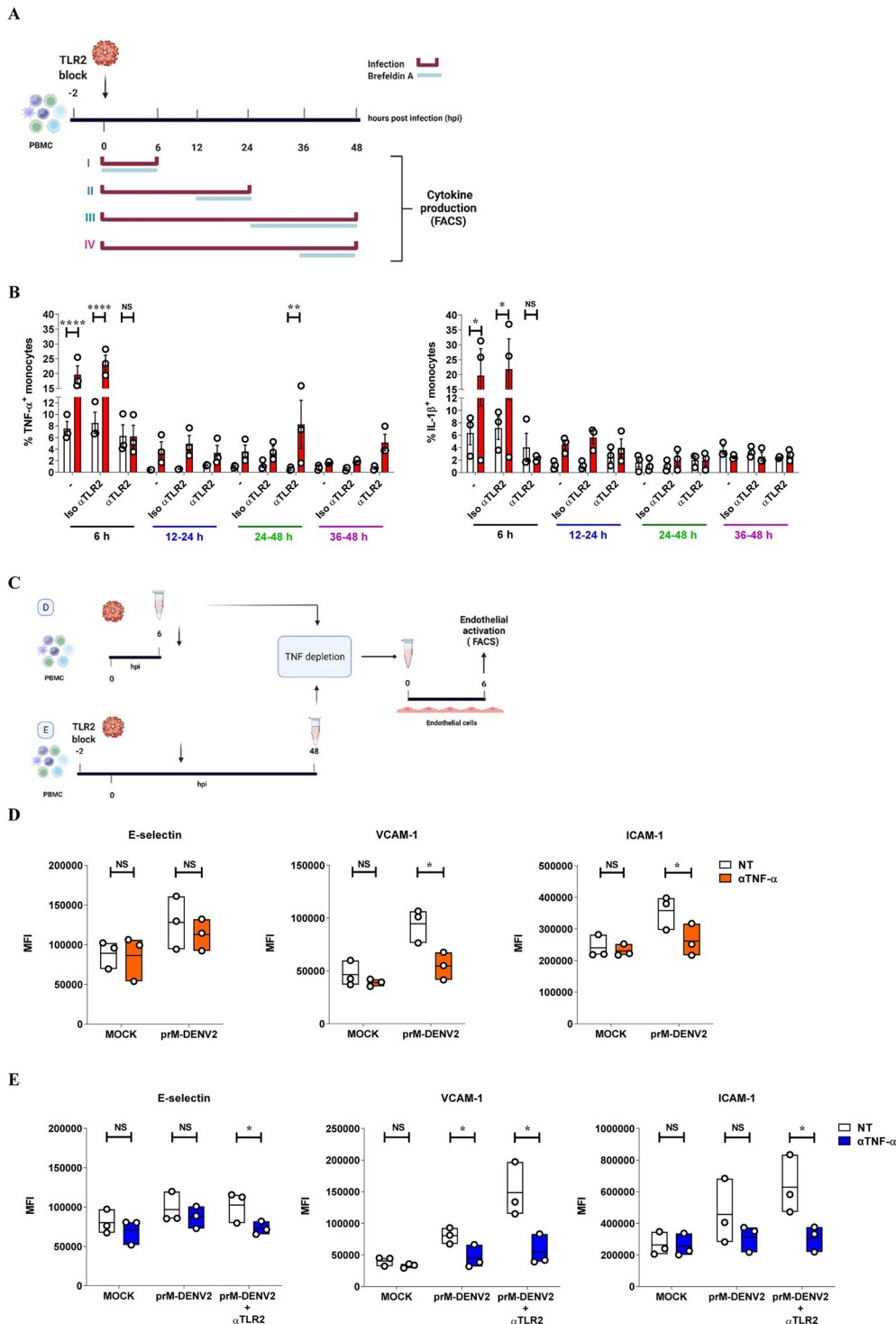

**Fig 7. Prolonged TLR2 block induces the release of TNF-α by monocytes exposed to prM-DENV2 and drives EC activation.** (**A**) Experimental scheme: hPBMCs (n = 3, three different donors) were treated with anti-TLR2 or TLR2 isotype control antibody (5 μg/mL) for 2 h prior exposure to prM-DENV2 (MOG 300) for 6h, 24h, and 48hpi in the presence of Brefeldin-A as indicated in the scheme. (**B**) Percentage of monocytes (in PBMCs) with intracellular expression of TNF-α and IL-1β was measured by flow cytometry. (**C**) Experimental scheme: supernatants from PBMCs (n = 3, three different donors) exposed to prM-DENV2 for 6h and 48 hpi, were incubated for 1h in the presence or absence of anti-TNF-α antibody (4 μg/ml). PBMCs exposed to prM-DENV2 for 48h were treated with anti-TLR2 2h prior prM-DENV2 stimulation. TNF-α-depleted PBMCs supernatants were used to stimulate HUVEC

for (**D**) 6 hpi and (**E**) 48hpi. Surface expression of E-selectin, VCAM-1 and ICAM-1was determined by flow cytometry and represented as MFI. (**B**) Bars represents mean± SEM. P values were obtained by two-way ANOVA with Tukey's post-test or unpaired one-tailed t-test (NS: not significant; *P<0.05; **P<0.01; ***P<0.001; ****P<0.0001). (**A** and **C**) Figures created with BioRender.com.

Immature DENV2 induced an early onset of soluble inflammatory responses in human PBMCs which in turn activated ECs in a TLR2/TLR6/CD14 dependent manner. Interestingly, however, the prolonged exposure of PBMCs to prM-DENV2 resulted in a significant decrease of TLR2 surface expression on monocytes. Additionally, the presence of TLR2/TLR6/CD14 blocking antibodies resulted in an increased production of soluble inflammatory mediators by PBMCs. We speculate that the decrease of monocyte TLR2 surface expression may reflect tolerance towards immature particles, which is supported by the resolution of the soluble inflammatory responses at 48h-72h. Previous studies have shown the association between decreased surface levels of TLR4 in murine macrophages and tolerance to endotoxin [56]. In addition, Butcher *et al.*, performed a comparative transcriptomic analysis of murine macrophages tolerized with TLR2, TLR3, TLR4 and TLR9 ligands and observed that TLR-induced tolerance generally shifts towards an anti-inflammatory response and concluded that the differential repression of inflammatory mediators is specific to the TLR ligand [57]. Nevertheless, tolerance does not explain how the presence of TLR blocking antibodies exacerbate inflammatory responses upon long stimulation with TLR ligands. The clue may lie in the fact that while exposure to prM-DENV induced release of sTLR2 blocking TLR2 may have affected shedding of this decoy receptor. Considering the anti-inflammatory role of sTLR2 during inflammation [52], we are currently evaluating its contribution to the resolution of inflammation.

Immature dengue particles have been previously shown to contribute to the innate immune responses by triggering potent type I IFN responses [30]. In that study, Décembre *et al.*, co-cultured pDCs isolated from blood with immature DENV-producing cells and observed an increase in the production of IFN-α by pDCs. Here, however, we could not detect IFN-α production in response to prM-DENV. The different cytokine responses observed in our study and that of Décembre's may be attributed to the different experimental approaches employed,

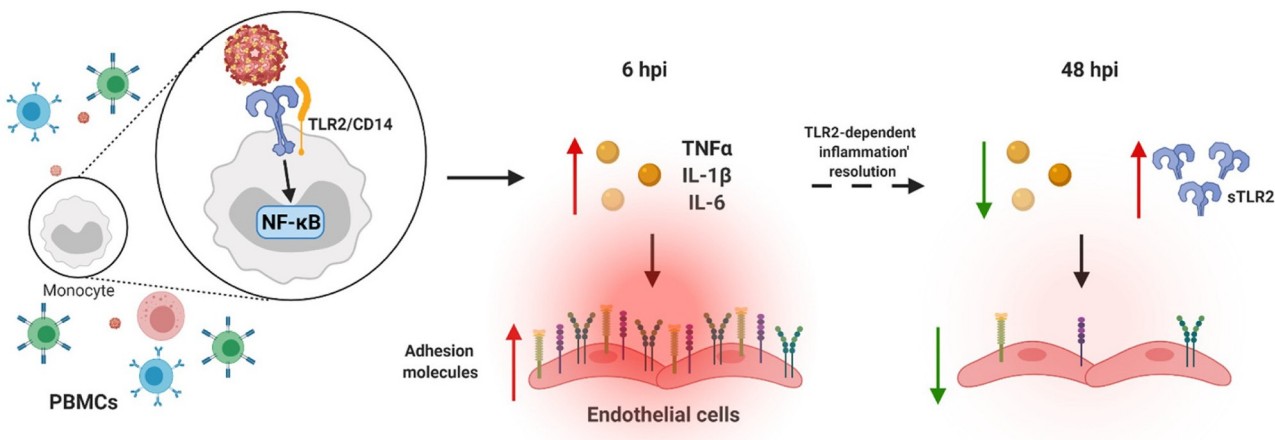

**Fig 8. Proposed model of immature DENV sensing by TLR2.** Sensing of prM-DENV2 by monocytes (within PBMCs) induce an early onset (6h) of soluble inflammatory responses leading to the activation of endothelial cells, as evidence by the upregulation of adhesion molecules E-selectin, VCAM-1 and ICAM-1 on endothelial cells (ECs). The production of inflammatory mediators including TNF-α, IL-1β and IL-6 was in control of TLR2 and its co-receptors TLR6 and CD14 and relies on the activation of the transcription factor NF-κB. Importantly, at later time points (48hpi) TLR2 axis functions to dampen the inflammatory responses as evidenced by downregulation of adhesion molecules on ECs. Figure created with BioRender.com.

enriched DC cultures vs PBMCs, in which pDCs represent less than 1% of all cells. It is worth mentioning that using the same infection model as here, we did detect IFN-type I responses in response to standard and infectious DENV preparation[17]. It is thus tempting to speculate that the noted difference is driven by the very inefficient entry, and infection of the immature DENV in PBMCs vs in pDCs enriched population, as consequence, the differential engagement of intracellular pattern recognition receptors sensing viral RNA and replication intermediates e.g., TLR3, TLR7, TLR8, RIG-I and MDA5 [12–14]. It has been previously shown that immature DENV virus can enter myeloid dendritic cells via DC-SIGN and initiate low grade infection [29]. Whether prM-DENV entry and RNA-sensing is also required for pDCs remains elusive. Also, in our PBMCs model, single TLR/CD14 treatments were studied. Therefore during the presence of TLR2 block, other TLR2 co-receptors such as CD14 and CD36 that have been shown to induce signaling independent of TLR engagement [58,59], may be sensing prM-DENV. Alternatively, C-type lectins present on myeloid cells could also play a role in the sensing [60]. Further studies are required to evaluate the differential engagement and/or crosstalk of these PRR's on PBMCs upon exposure to immature DENV in the presence of TLR2 block.

DENV is known to hijack CME to gain access to the internal compartments of host cells [19]. In addition, TLR2 ligands LTA, FSL-1 and PAM3CSK4 are internalized via CME in which TLR2 co-receptors: CD14 and CD36 play a very important role [37,39]. In the current study, we questioned if the sensing of immature DENV particles occurred at the plasma membrane and/or in an endosomal compartment. We found that internalization of immature DENV virions by the TLR2/TLR6/CD14 axis via CME potentiates subsequent NF-κB activation. These data are in line with previous findings by us and others [17,37]. However, the fate of immature DENV virions following internalization via CME is unknown. Triantafilou et al., showed that TLR2, CD14 and CD36 reside in lipid rafts, following ligand binding TLR1 and TLR6 are recruited to lipid rafts and TLR2 clusters are targeted to the Golgi apparatus. Localization and targeting inside the cell are ligand-specific, being trafficking independent of signaling [61]. Subsequent studies should elucidate the fate of immature DENV following sensing via TLR2/TLR6/CD14 and explore the role of the TLR2/TLR6 co-receptor CD36 on signaling, trafficking and internalization.

We and others have previously demonstrated that antibodies against the E and prM glycoproteins can rescue the infectious properties of immature DENV by facilitating the binding and opsonization of these virions to Fcγ-receptor expressing cells, a phenomenon known as antibody-dependent enhancement (ADE) [28,62,63]. Our current study did not address the role of infection-enhancing antibodies in the TLR2-mediated immune responses induced by prM-DENV. Consequently, future studies should address the inflammatory responses triggered by immature virions in the course of secondary infections.

In summary, our data uncover the important role of TLR2 and its co-receptors, TLR6 and CD14, in the regulation of the early and late soluble inflammatory responses to immature DENV particles. Our findings indicate that presence of immature virions in blood may modulate the kinetics of inflammation and thereby contribute to dengue virus disease pathogenesis.

## Material and methods

### Ethics statement

Buffy coats from Sanquin Blood Bank were used to isolate PBMCs after written informed consent by the donors. All procedures were approved by The Sanquin Ethical Advisory Council and are in accordance with the declaration of Helsinki and Dutch regulations.

## Cells

Human adenocarcinoma active furin-deficient LoVo cells were cultured in Ham's medium (Invitrogen) supplemented with 20% FBS. *Aedes albopictus* C6/36 cells were cultured in MEM (Life Technologies) supplemented with 10% FBS, HEPES (25 mM), sodium bicarbonate (7.5%), penicillin (100 U/mL), streptomycin, glutamine (100 μg/mL) and non-essential amino acids. Baby hamster kidney cells clone 15 (BHK-15) were maintained in DMEM supplemented with 10% FBS, penicillin (100 U/mL), streptomycin (100 μg/mL), non-essential amino acids (100 μM) and HEPES (10 mM). HEK-Blue hTLR2 and HEK-Blue Null1 cells (InvivoGen) were cultured in DMEM supplemented with 10% FBS, penicillin (100 U/mL), streptomycin (100 μg/mL) and maintained according to the manufacturer's instructions. Buffy coats obtained from anonymized, DENV-seronegative, Sanquin blood donors were used to isolate PBMCs. The PBMCs were cryopreserved at -196 ˚C. Primary human umbilical vein endothelial cells (HUVEC) (Lonza, the Netherlands) were cultured in EBM-2 supplemented with EGM-2 endothelial growth SingleQuot kit supplement & growth factors (Lonza, the Netherlands). All of the cell lines used in this study tested negative for the presence of Mycoplasma spp. using a commercial functional method (Lonza, the Netherlands) and/or in-house qPCR assay adapted from Baronti *et al* [64].

## Viruses

Standard (std) DENV1 (strain 16007), DENV2 (strain 16681) and DENV4 (strain 1036) were produced in the *Aedes albopictus* C6/36 cell line as described before [28]. Briefly, an 80% confluent monolayer of cells was infected at multiplicity of infection (MOI) of 0.1. Virus progeny was harvested at 72 or 168 hours post infection (hpi) depending on the serotype. prM-DENV1 (strain 16007), prM-DENV2 (strain 16681), and prM-DENV4 (strain 1036), were produced in LoVo cells as described [27,29]. Briefly, an 80% confluent monolayer of cells was infected at multiplicity of infection (MOI) of 2 and 72hpi the immature preparations were harvested. The cleavage of prM into M does not occur as LoVo cells are deficient of the protease furin, therefore the secreted particles are fully immature [22,27]. Viral preparations were analyzed with respect to the infectious titer and the number of genome equivalents copies. The infectivity of immature DENV was determined by measuring the number of plaque-forming units (PFU) by plaque assay on BHK-15 cells and the number of genome-equivalent copies (GEc) by quantitative RT-PCR (RT-qPCR), as described previously [27,65]. In line with previous findings [27], the specific infectivity of prM-DENV preparations is on average 10,000 lower than that of std DENV preparation with GEC to PFU ratio of $10^6$ vs std DENV2 between $5x10^1$-$5x10^2$. Virus inactivation was performed by 1h incubation of virus aliquots under UVS-28 8-watt Lamp. Inactivation to below level of detection 35 PFU/mL was confirmed using standard plaque assay on BHK-15 cells as described previously. All virus preparations used in this study were tested negative for Mycoplasma spp. using a commercial functional method (Lonza, the Netherlands) and/or in-house qPCR assay adapted from Baronti *et al* [64].

## In vitro treatments

PBMCs ($1x10^6$ cell/mL, 48-well plate) or HEKs ($5X10^4$ cells/mL, 96-well plate) cells were treated with 5 μg/mL or 15 μg/mL, respectively of anti-hTLR2-IgA (clone B4H2), pab-hTLR6-IgG (polyclonal), pab-hTLR1-IgG (polyclonal), anti-hCD14-IgA (2.5 μg/mL (PBMCs), clone D3B8), human IgA2 isotype control (clone T9C6), normal rat PAb IgG control blocking antibodies (InvivoGen), Dexamethasone (10μM, InvivoGen), Bay-11 (5μM, InvivoGen) or MG-132 (9.5μM, InvivoGen), C29 (100μM, MedChemExpress) for 2h followed by treatment with PAM3CSK4 (600 ng/mL (PBMCs), 50 ng/mL (HEK's), InvivoGen), prM-DENV1 (MOG

250, MOG 300), prM-DENV2 (MOG 250, MOG 300, MOG 1000), prM-DENV4 (MOG 250, MOG 300, MOG 1000), UV-I prM-DENV2 (MOG 300) and std DENV2 (MOG 300) for 6h, 12h, 24h, 48h and 72h. For viral production analysis, supernatants from 48h and 72h later supernatants were collected and preserved at -80˚C. For endothelial cells activation and cytokine production analysis, cell-free supernatants were collected at 6h, 12h, 24h, 48h and 72h post treatment and preserved at -80˚C.

## Endothelial cell activation assay

HUVEC were incubated with cell-free supernatants from prM-DENV-treated PBMCs. In addition, direct exposure to prM-DENV2, RPMI media and TLR4 agonists LPS, served as controls. After 6h of stimulation, cells were used directly for the expression analysis of activation markers using FACS as described. Briefly, cells were stained with the following antibodies CD62E E-selectin PE (clone HCD62E), CD106 VCAM-1 APC (clone STA) and CD54 ICAM-1 FITC (clone HCD54). Isotype-matched controls labelled with PE (clone RMG2a-62), APC (clone RMG1-1) and FITC (clone RMG1-1) were used as negative controls for comparing E-selectin, VCAM-1 and ICAM-1 expression, respectively. All surface staining antibodies were from BioLegend. Data were collected using a FACSverse flow cytometer (BD Biosciences) and analyzed using the Flowjo (BD Biosciences).

## NF-κB reporter assay

Human embryonic kidney reporter cell lines (HEK-Blue hTLR2; $5 \times 10^4$ cells/well) were infected or treated as described in previous sections in a 96-well flat bottom plate for 24h at 37˚C, 5% CO2 atmosphere. To test if endocytosis is required for NF-κB stimulation, wortmannin (Sigma Aldrich), pitstop 2 (Sigma Aldrich), dynasore (Sigma Aldrich) or ammonium chloride ($NH_4Cl$, Merck) were added 1h before infection or treatment with the agonist. After 24h post infection or treatment, analysis of NF-κB stimulation was assessed by mixing the supernatants with QUANTI-Blue (InvivoGen). Absorbance was measured at 630 nm using Synergy HT multi-mode microplate reader (BIOTEK). PAM3CSK4 (50 ng/mL) was used as a positive control.

## Cytokine and chemokine determination

Human anti-virus response panel (13-plex, LEGENDplex, BioLegend) was used to determine the protein levels of IL-1β, TNF-α, IL-6, IL-8, IL-10, IL-12p70, IP-10, GM-CSF, IFN-α2, IFN-β, IFN-γ, IFN-λ1 and IFN-λ2/3. Data were collected using a FACSverse flow cytometer (BD Biosciences) and analyzed using LEGENDplex v8.0 (BioLegend).

## Intracellular cytokine detection and TLR2 surface expression

Intracellular accumulation of IL-1β and TNF-α was measured at 6h, 12h, 12-24h, 36-48h and 24-48h post treatment. Briefly, Brefeldin A (10 μg/mL, BioLegend) was added immediately after treatments or in the last 12 hours of cell stimulation. PBMCs were then incubated with fixable viability dye eFluor 780. Cells were fixed, permeabilized and stained with TNF-α eFluor 450 (clone MAb11) or IL-1β FITC (Clone B-A15), from eBioscience. Surface expression of TLR2 was measured on PBMCs at 48h post treatment. Briefly, PBMCs were incubated with fixable viability dye eFluor 780, CD282 (TLR2) PE (clone TLR2.1), CD14 eFluor 450 (Clone 613D) and CD16 APC (clone CB16), all from eBioscience. PBMCs were fixed and washed with FACS buffer (PBS 1X, 2% FBS). Samples were measured on a FACSverse flow cytometer (BD Biosciences) or NovoCyte Quanteon cytometer (Agilent, CA, United States). Isotype-matched

antibodies labelled with PE (clone EBM2a), eFluor 450 (clone P3.6.2.8.1) APC (clone P3.6.2.8.1) from eBioscience and FITC (IgG1, Invitrogen) were used as negative controls to compare the expression of the respective marker. Data were analyzed using the FlowJo software (version 10, Tree Star, San Carlo, CA).

## TNF-α depletion on PBMCs supernatants

For the TNF-α depletion assays we adapted a previously described protocol[53]. Briefly, supernatants from (mock-) prM-DENV2- treated PBMCs were treated (1h, 37 ˚C) in the presence or absence of anti-TNF-α antibody (4 µg/mL, BioLegend). The TNF-α depleted supernatants were used to stimulate HUVEC for 6 hours and the expression of activation markers were analyzed by flow cytometry. Recombinant TNF-α (0.1–10 ng/ml, BioSource, the Netherlands) was used as a positive control. Data were collected using a FACSverse flow cytometer (BD Biosciences) and analyzed using the FlowJo (BD Biosciences).

## Analysis of prM-DENV infectivity

BHK-15 cells were incubated with supernatants from prM-DENV2-treated PBMCs. DENV infection was then measured by intracellular detection of the E protein using the murine monoclonal 4G2 antibody (Millipore) followed by rabbit anti-mouse IgG-coupled to AF647 (Molecular probes). Unstained cells, mock-treated cells plus secondary antibody only, mock-treated cells plus detection pair and prM-treated cells plus secondary antibody were included as controls. In addition, Std DENV2 and UV-I prM-DENV2 were used as positive and negative controls for infection, respectively. Samples were measured on a FACSverse cytometer (BD Biosciences) and data were analyzed with Flowjo (BD Biosciences). HUVEC were exposed to prM-DENV, std-DENV, or their UV-inactivated equivalent at MOG 230 for 24h, 48h, and 72 hp. DENV infection was measured by detection of the NS3 protein via western blot using mouse anti-DENV2 NS3 (ThermoFisher) and mouse anti-GAPDH (Abcam) and HRP-conjugated goat anti-mouse (Merck). Pierce ECL western blotting substrate (ThermoFisher) or SuperSignal West FEMTO (ThermoFisher) was used for detection by chemiluminescence using LAS-400 mini camera system (GE-Healthcare). Alternatively, infection was determined via immunofluorescence staining. Briefly, cells were fixed for 24h, permeabilized and stained with mouse anti-flavivirus E-protein (Millipore) followed by rabbit anti-mouse (ThermoFisher). Slides were mounted with Pro-Long Gold Antifade with DAPI (ThermoFisher) and analyzed with Leica DM4000 B (Leica Microsystems).

## Statistical analysis

Data analysis was performed using Prism 9.3.0 (Graphpad, USA). Unless indicated otherwise, data are shown as media ± SD or SEM. One-way ANOVA followed by Dunnet's post- hoc test was used for comparisons vs experimental controls. Two-way ANOVA with Tukey's post- hoc test was used for time-point comparisons. Paired one-tailed t-test or unpaired one-tailed t-test were used to determine statistical significance of single experimental results. In all tests, values of $^*p<0.05$, $^{**}p<0.01$, $^{***}p<0.001$ and $^{****}p<0.0001$ were considered significant.

## Supporting information

**S1 Fig. Direct exposure of HUVEC to prM-DENV2 does not lead to upregulation of adhesion molecules.** HUVEC were (mock) treated with LPS (1µg/mL) or prM-DENV2 (MOG 300) for 6h. Surface expression of E-selectin, VCAM-1 and ICAM-1 was determined by flow cytometry. (**A**) Gating strategy on HUVEC to define adhesion molecule positive cells. (**B**)

Expression of adhesion molecules following direct stimuli with prM-DENV2 and LPS. Bar represents mean± SD of two independent biological experiments.
(PDF)

**S2 Fig. prM-DENV2 induces an early onset of soluble inflammatory responses.** (**A**) HUVEC were incubated with cell-free supernatants harvested at indicated time points from PBMCs (n = 3, three different donors) exposed to prM-DENV2 at an MOG of 300 or mock treatment. Surface expression of E-selectin, VCAM-1 and ICAM-1 on HUVEC was determined by flow cytometry and represented as (**B**) percentage of positive cells and (**C**) MFI. D denotes PBMC donor.
(PDF)

**S3 Fig. Immature DENV2 is not infectious in human PBMCs and HUVEC. (A)** Baby Hamster Kidney cells clone 15 (BHK-15) were incubated for 24h with standard (std) DENV at an MOI of 5 (positive control) or cell-free supernatants from PBMCs (n = 2, two different donors) exposed to prM-DENV2 (MOG 300), UV-prM-DENV2 (MOG 300) or mock treatment for 48h and 72h. % of DENV E+ cells was determined by flow cytometry. (**B**) HUVEC were infected with Std DENV2 (MOI 20) and prM DENV2 (MOG 10 000) for 24h, cells were then stained for flavivirus E-protein (n = 1). (**C**) HUVEC were infected with prM-DENV2 and Std DENV2 and their respective UV-inactivated preparations at MOG 230 for 24h, 48h and 72 h. Detection of non-structural protein 3 (NS3) was detected by Western blot. GAPDH was used as a reference. Representative blot and Quantification of NS3 expression shown as fold change to the respective GAPDH control (n = 1).
(PDF)

**S4 Fig. Sensing of immature dengue virions is TLR2 specific.** (**A**) HEK-Blue Null1 cells (mock)-treated with PAM3CSK4 (PAM3, 50 ng/mL, n = 2), prM-DENV2 (MOG 300, n = 2) for 24h. (**B**) HEK-Blue hTLR2 cells were (mock)-treated with pure (n = 2) or crude (n = 3) preparations of prM-DENV2 (MOG 250) for 24h. (**C**) HEK-Blue hTLR2 cells (mock)-treated with PAM3CSK4 (PAM3, 50 ng/mL, n = 3), prM-DENV1 (MOG 250, n = 1) and prM-DENV4 (MOG 250, n = 1; MOG 1000, n = 2) for 24h. NF-κB stimulation was assessed by QUANTI-Blue, OD values show the induction of NF-κB. Data represents the mean ± SEM. P values were obtained by one-way ANOVA, Dunnett post hoc test (****$P<0.0001$). n = independent biological experiments. CC: cellular control.
(PDF)

**S5 Fig. Isotype controls do not impair the TLR2-dependent prM-DENV2-induced production of cytokines.** PBMCs from healthy donors (n = 2–3, two to three different donors) were mock-treated with αTLR2 and αTLR1/TLR6 isotype control antibodies (5 μg/mL) for 2 hours prior exposure with prM-DENV2 at MOG 300 for 6h. Cytokine production was measured by flow cytometry using LegendPlex. Each graph shows the production of each cytokine in picograms per milliliter (pg/mL).
(PDF)

**S6 Fig. prM-DENV2-induced cytokines independently of TLR2/TLR6/CD14 axis.** PBMCs from healthy donors (n = 2, two different donors) were mock-treated with αTLR2, αTLR1, αTLR6 (5 μg/mL), αCD14 (3 μg/mL) and isotype control antibodies (5 μg/mL) for 2 hours prior exposure to prM-DENV2 at MOG 300 for 6h. Cytokine production was measured by flow cytometry using LegendPlex. Each graph shows the production of each cytokine in picograms per milliliter (pg/mL).
(PDF)

**S7 Fig. Monocytes within the PBMCs are the main source of TNF-α and IL-1β after exposure to prM-DENV2.** PBMCs from healthy donors (n = 3–5, three to five different donors) were mock-treated with αTLR2 (5 μg/mL), αTLR1 (5 μg/mL), αTLR6 (5 μg/mL), αCD14 (3 μg/mL), αTLR2 and αTLR1/TLR6 isotype control antibodies (5 μg/mL), Dexamethasone (DEX, 10μM), Bay-11 (5μM) and MG-132 (9.5 μg/mL) for 2 hours prior exposure to prM-DENV2 at MOG 300 or PAM3CSK4 (PAM3, 600 ng/mL) for 12h in the presence of Brefeldin-A. (**A**) Gating strategy to measure the intracellular accumulation of IL-1β and TNF-α in the live monocyte and lymphocyte fractions within the PBMCs. (**B, C** and **F**) Intracellular accumulation of IL-1β and TNF-α in live lymphocytes. (**D** and **E**) percentage of viable PBMCs, monocytes, and lymphocytes. Bar represents the mean ± SEM.
(PDF)

**S8 Fig. Pharmacological targeting of TLR2 in hPBMCs abrogates inflammatory responses induced by PAM3.** (**A**) PBMCs from a healthy donor (n = 1, one donor) were mock-treated with C29 (100 μM), vehicle control DMSO and PAM3CSK4 (PAM3, 600 ng/mL) for 6h, 24h and 48h. Percentage of viable PBMCs was measured by flow cytometry. (**B** and **C**) PBMCs from a healthy donor (n = 1, one donor) were mock-treated with C29 (100 μM) or vehicle control DMSO for 2h prior exposure to PAM3CSK4 (PAM3, 600 ng/mL) for 6h in the presence of Brefeldin-A. (**B**) Intracellular accumulation of TNF-α, IL-1β and IL-6 in the live monocyte fraction within the PBMCs was measured by flow cytometry. (**C**) HUVEC were mock-treated with C29 (100 μM) or vehicle control DMSO for 6h. Surface expression of E-selectin, VCAM-1 and ICAM-1 on HUVEC was determined by flow cytometry.
(PDF)

**S9 Fig. Pharmacological targeting of TLR2 in hPBMCs abrogates inflammatory responses induced by prM-DENV2.** PBMCs from healthy donors (n = 3, three different donors) were mock-treated with C29 or DMSO (vehicle) for 2h prior exposure to prM-DENV2 (MOG 300) or mock treatment for 6h. Cell-free supernatants were collected and used to stimulate HUVEC. Surface expression of E-selectin, VCAM-1 and ICAM-1 on HUVEC was determined by flow cytometry and represented as (**A-C**) percentage of adhesion molecule positive cells and (**E**) MFI fold change to non-treated control. P values were obtained by unpaired one-tailed t-test ($^{*}$P<0.05; $^{**}$ P<0.01; $^{***}$P<0.001).
(PDF)

**S10 Fig. prM-DENV2-induced production of soluble inflammatory mediators is TLR2 specific.** HUVEC were incubated for 6h with cell-free supernatants from PBMCs (n = 2–3, two to three different donors) (mock)-exposed for (**A**) 6h and (**B**) 48h to prM-DENV2 at an MOG of 300 in the presence or absence of TLR2 isotype control antibody (5 μg/mL). Surface expression of E-selectin, VCAM-1 and ICAM-1was determined by flow cytometry and represented as percentage of positive cells and MFI. Orange: 6h, Blue: 48h.
(PDF)

**S11 Fig. Early onset of soluble inflammatory responses induced by prM-DENV1 and prM-DENV4 are in control of TLR2/TLR6/CD14 axis.** HUVEC were incubated for 6h with cell-free supernatants from PBMCs (n = 2, two different donors) exposed for 6h to (**A**) prM-DENV1 and (**B**) prM-DENV4 at an MOG of 300 in the presence or absence of TLR2 isotype control antibody (5 μg/mL). Surface expression of E-selectin, VCAM-1 and ICAM-1was determined by flow cytometry and represented as percentage of positive cells and MFI.
(PDF)

**S12 Fig. TLR2 block enhances the late onset of soluble inflammatory responses induced by prM-DENV1 and prM-DENV4.** HUVEC were incubated for 6h with cell-free supernatants from PBMCs (n = 2, two different donors) exposed for 48h to (**A**) prM-DENV1 and (**B**) prM-DENV4 at an MOG of 300 in the presence or absence of TLR2 isotype control antibody (5 μg/mL). Surface expression of E-selectin, VCAM-1 and ICAM-1was determined by flow cytometry and represented as percentage of positive cells and MFI.
(PDF)

**S13 Fig. Prolonged TLR2 block does not induce the intracellular accumulation of TNF-α and IL-1β in lymphocytes (within PBMCs) exposed to prM-DENV2.** hPBMCs (n = 3, three different donors) were treated with anti-TLR2 or TLR2 isotype control antibody (5 μg/mL) for 2h prior exposure to prM-DENV2 (MOG 300) for 6h, 24h and 48h in the presence of Brefeldin-A. The intracellular accumulation of (**A**) TNF-α and (**B**) IL-1β was measured in lymphocytes (within PBMCs) by flow cytometry. Bar represents the mean ± SEM.
(PDF)

**S14 Fig. Depletion of TNF-α reduces the TNF-α-mediated HUVEC activation.** Different concentrations of recombinant TNF-α (rTNF-α, 0.1–10 ng/ml) were incubated in the presence or absence of anti-TNF-α antibody (4 μg/ml) for 1 hour. HUVEC were then stimulated with the rTNF-α/ anti-TNF-α preparations for 6h and the surface expression of E-selectin, VCAM-1 and ICAM-1 was determined by flow cytometry and showed as MFI normalized to relative mock values. Bar represents the mean ± SEM.
(PDF)

## Acknowledgments

We thank Jorge Andres Castillo Ramirez, Heidi van der Ende-Metselaar, Jelmer Niewold and Anna van der Sluis for their technical assistance. We are grateful to Henk Moorlag and Timara Kuiper (Endothelial Biomedicine & Vascular Drug Targeting group, UMCG) for his technical assistance in the culture of the HUVEC. We thank Geert Mesander (Flow Cytometry Unit, UMCG) for his technical expertise and guidance in the flow cytometry assays.

## Author Contributions

**Conceptualization:** José Alberto Aguilar Briseño, Lennon Ramos Pereira, Jill Moser, Izabela A. Rodenhuis-Zybert.

**Data curation:** José Alberto Aguilar Briseño, Lennon Ramos Pereira.

**Formal analysis:** José Alberto Aguilar Briseño, Lennon Ramos Pereira, Marleen van der Laan, Mindaugas Pauzuolis, Bram M. ter Ellen, Vinit Upasani.

**Funding acquisition:** José Alberto Aguilar Briseño, Izabela A. Rodenhuis-Zybert.

**Investigation:** José Alberto Aguilar Briseño, Lennon Ramos Pereira, Marleen van der Laan, Mindaugas Pauzuolis, Bram M. ter Ellen, Vinit Upasani.

**Methodology:** José Alberto Aguilar Briseño, Lennon Ramos Pereira, Jill Moser, Izabela A. Rodenhuis-Zybert.

**Project administration:** José Alberto Aguilar Briseño, Izabela A. Rodenhuis-Zybert.

**Resources:** José Alberto Aguilar Briseño, Izabela A. Rodenhuis-Zybert.

**Supervision:** José Alberto Aguilar Briseño, Izabela A. Rodenhuis-Zybert.

**Validation:** José Alberto Aguilar Briseño, Lennon Ramos Pereira, Marleen van der Laan, Mindaugas Pauzuolis, Bram M. ter Ellen, Vinit Upasani.

**Visualization:** José Alberto Aguilar Briseño, Lennon Ramos Pereira, Marleen van der Laan, Vinit Upasani, Izabela A. Rodenhuis-Zybert.

**Writing – original draft:** José Alberto Aguilar Briseño, Lennon Ramos Pereira, Izabela A. Rodenhuis-Zybert.

**Writing – review & editing:** José Alberto Aguilar Briseño, Lennon Ramos Pereira, Marleen van der Laan, Mindaugas Pauzuolis, Bram M. ter Ellen, Vinit Upasani, Jill Moser, Luís Carlos de Souza Ferreira, Jolanda M. Smit, Izabela A. Rodenhuis-Zybert.

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
