## [Decision Letter · Decision Letter 0]

19 May 2022

Dear Dr. Rodenhuis-Zybert,

Thank you very much for submitting your manuscript "TLR2 axis on peripheral blood mononuclear cells regulates inflammatory responses to circulating, non-infectious immature dengue virus particles" for consideration at PLOS Pathogens. As with all papers reviewed by the journal, your manuscript was reviewed by members of the editorial board and by several independent reviewers. In light of the reviews (below this email), we would like to invite the resubmission of a significantly-revised version that takes into account the reviewers' comments.

We cannot make any decision about publication until we have seen the revised manuscript and your response to the reviewers' comments. Your revised manuscript is also likely to be sent to reviewers for further evaluation.

Sincerely,

Alain Kohl

Associate Editor

PLOS Pathogens

Ana Fernandez-Sesma

Section Editor

PLOS Pathogens

Kasturi Haldar

Editor-in-Chief

PLOS Pathogens

orcid.org/0000-0001-5065-158X

Michael Malim

Editor-in-Chief

PLOS Pathogens

orcid.org/0000-0002-7699-2064

Reviewer's Responses to Questions

**Part I - Summary**

Reviewer #1: Dengue (caused by dengue virus DENV) is a major clinical problem without licensed vaccines nor antivirals and so understanding host-virus interactions with regard sensing and inflammation is important and my help design novel safe and effective interventions.

Aguilar-Briseño et al., here describe a follow up study from their 2020 paper linking investigating link between TLR2 DENV sensing and pathogenesis. In their previous study, immature particles were not assessed, which is where is the main novelty of this new work lies. Immature, poorly processed virions are commonly found in vivo and their biological role is poorly understood. Their work across both papers highlights the important but complex role of TLR2 sensing during DENV infection in that acute or chronic stimulation results in distinct outcomes.

Aguilar-Briseño et al use a two-cell model of DENV host interactions looking at DENV challenge of PBMCs or pure cell populations followed by incubation of conditioned media with primary vascular cells, which are heavily involved in DENV pathogenesis. To assess the contribution of immature particles, they produce fully immature virions by infection of furin-defieicnt LoVo cell lines. Additional systems looked at include various HEK reporter lines. Using these approaches, the team show immature virions are sensed by PBMCs and monocytes (and HEK cells) in a TLR2(and coreceptor TLR6/CD14)-dependent manner, which results in production of secretable factors that can activate endothelial cells. Interestingly, prolonged TLR2 activation/DENV particle incubation results in enhanced inflammatory mediator production and vascular activation, possibly linked to TNFa.

This work forms the basis of further studies exploring production of DENV immature virions and kinetics of host immune response during primary and secondary infections. It still remains uncertain whether TLR2 sensing is protective or pathogenic across the spectrum of DENV infection in people.

In general, this is a very well-written and produced study, building on - and extending - solid previous work in a scientifically and clinical relevant arena. This reviewer appreciated the ease of presentation, inclusion of various additional controls and mechanistic focus. The stats appear to be carried out appropriately. The study could have benefited from greater numbers of PBMC donors per experiment, genetic KO models of TLR2 in primary monocytes. However, where appropriate, statistical significance and magnitude of change in primary cells is clear, and combination of neutralizing antibodies and HEK cell reporters allowed mechanistic insight. I have several minor points that would help aid in interpretation of the work.

Reviewer #2: In this well developed manuscript, Aguilar-Briseño and colleagues demonstrate that TLR2 in PBMCs mediates an inflammatory response to immature DENV particles. Using DENV with a significant amount of prM on its surface (prM-DENV), they show that TLR2 signaling is induced in PBMCs exposed to this virus, though not directly infected. Through pharmacological inhibition of distinct steps in the pathway, they further show that NFkB signaling and downstream cytokine leads to endothelial cell responses to infection. Finally, they nicely show that these responses are downregulated through TLR2 shedding. Overall, the manuscript is well written and understandable, the figures are nicely described and the diagrams for experiments and the model are very appreciated, and the experiments are logical.

Reviewer #3: This study is very similar to the author’s prior work which showed that TLR2 on monocytes is activated by circulating dengue virus (PMID 32576819). Previously the authors had not specified whether mature or immature virions can trigger this pathway. Here they show TLR activation of PBMCs by immature DENV and characterize the down-stream signaling that follows TLR2 activation, which is an already well characterized pathway. Like their previous study, they again show that monocytes are the main responders in PBMCs to stimulation by virus (here immature virus). The proportions of mature and immature virus in vivo are not known so it is not clear how consequential this distinction is for physiological function.

**Part II – Major Issues: Key Experiments Required for Acceptance**

Reviewer #1: One significant concern is the choice of controls across the experiments and the worth of additional variables in interpretation of their results. i) Were the prMDENV preparations purified by ultracentrigfution or are they crude? How would this affect results? Ii) Would additional controls such as media from uninfected LoVo cells or from heat-treated prmDENV from LoVo be helpful? Iii) Furthermore, use of standard mature infectious DENV in some experiments would be useful to compare. Can the authors comment on the merits of these additional controls.

Reviewer #2: The authors primary motivation for the manuscript is the detection of prM-DENV via TLR2, which they demonstrate in Figure 1. They do not compare this response to a response from DENV infection, with less prM or with any other distribution of prM. Because the authors spend significant parts of the introduction and a full paragraph of the discussion describing the viral system, it would benefit the manuscript to have this comparison within the manuscript.

Their data regarding TLR2 shedding shows that samples prior to 48 hpi largely maintain TLR2 (no detectable sTLR2). Do the authors envision that the shedding of TLR2 is gradual? Additional timepoints between 24h and 48h would be informative.

It is surprising that the authors find little interferon in their samples, despite clear prM-DENV detection by the PBMCs. They attribute this to differences in the experimental systems (though their cultures also have DCs) or differences in the virus. Their suggestion that efficient infection leads to detection and IFN production is testable within their system, and would significantly benefit this statement.

Reviewer #3: There is no comparison between mature and immature triggered responses so we cant know if the central claim of the study that immature virus is uniquely triggering this pathway is actually true. This is a control missing throughout the study.

All of the studies presented were obtained using a culture system where supernatants are transferred from DENV-activated PBMCs to endothelial cells. There are no in vivo studies or validation that these responses occur in vivo

Given the fact that it has already been shown that DENV activates TLR2 signaling, most of the results of experiments performed validating cytokine production and NFkB activation etc. are fully expected.

I don’t think the flow cytometry based assay is sensitive enough to establish that there was not virus replication. Negative-strand PCR should be used. Also, this should be done on the HUVEC cells too, not only the PBMCs.

Growing virus using Lovo cells to prevent maturation could be used for some studies, but this is an artificial system and may not represent what actually occurs in DENV infection.

Although it is suggested that TLR2 shedding correlates with results in co-cultured endothelial cells, this was not explicitly shown.

**Part III – Minor Issues: Editorial and Data Presentation Modifications**

Reviewer #1: Title: remove reference to "circulating" as this refers to in vivo situation, which was not assessed in this study. Indeed, the short title is more accurate in my opinion.

Update author summary - reads very similar to abstract, which is not he object of author summary.

Intro:

Spell out RIGI/MDA5

Add in DENV RNA genome having positive sense polarity

Methods:

Cells section: add that LoVo cells lack furin in the first section listing LoVo (as well as in Viruses section)

Viruses section: is there meant to be a reference for production of prM-DENV in LoVo cells?

PMBCs: state what size well plate was used for your PBMC experiments

Were the same 3 donors used throughout?

When prm denv added, is it ever removed? What is its stability? Have the authors tried adding, removal and washing out the virions?

Are the tlr2 inhibitors in the conditioned media when added the vascular cells? Is it possible this could affect endothelial cell biology?

Often it is not clear the number of independent experiments nor what n refers to (wells or complete experiments on different days). Can the authors update each legend documenting this. Although noted everything seems consistent and no issues found.

The work would have benefited from more primary cell donors as significant variation is noted. On this, as variation is biologically meaningful, is there a possibility we can look at responses in PBMCs across experiments by donor or is it likely to reflect technical differences in experiments? Similarly, is it possible and relevant if these donors have had a previous DENV infection? Was this considered in use of donors?

Results:

Dengue virions : change to DENV virions

Alter: significant (> at least 2 fold) - does significant refer to statistics or to fold change magnitude?

Change the word "tagged" (as in tagged as std) to "referred to herein as".

Change dengue virus to DENV where applicable throughout

Re: role of tlr1, there does seem to be a trend towards lower activation upon its neutralization, although perhaps not statistically or biologically significant

Discussion:

What is sensing prmDENV immature in tlr2 block? Is there an additional sensor, possibly RNA or not, operating here during your setup?

What happens to the immature DENV upon tlr2 sensing? Are the particles internalised and removed?

Reviewer #2: Supplementary Figure 7 (or portions of it) seem rather important to the manuscript and it may benefit the paper to have some of these data in the main body of the text.

The difference in the response at 6h and 48h (Figure 5) is interesting and the authors nicely provide explanations for this difference. Their data would seem to indicate that blockage of TLR2/TLR6/CD14 enhances this late response. Are these data statistically significant?

Reviewer #3: (No Response)

PLOS authors have the option to publish the peer review history of their article (what does this mean?). If published, this will include your full peer review and any attached files.

Reviewer #1: **Yes: **Connor G G Bamford

Reviewer #2: No

Reviewer #3: No
---

## [Decision Letter · Decision Letter 1]

30 Aug 2022

Dear Dr. Rodenhuis-Zybert,

Thank you very much for submitting your manuscript "TLR2 axis on peripheral blood mononuclear cells regulates inflammatory responses to non-infectious immature dengue virus particles" for consideration at PLOS Pathogens. As with all papers reviewed by the journal, your manuscript was reviewed by members of the editorial board and by several independent reviewers. In light of the reviews (below this email), we would like to invite the resubmission of a significantly-revised version that takes into account the reviewers' comments.

While two of the reviewers were positive about the manuscript, the third reviewer still has substantial concerns following revision.

The authors should look at the all comments carefully, and: specifically address the concerns on the comparative analysis in figure 1; clarify the use of Lovo cells and their use; state the numbers of donors as used per experiment and discuss the variability observed; and verify or at least discuss the final comment on validating the lack of infection by qPCR.

We cannot make any decision about publication until we have seen the revised manuscript and your response to the reviewers' comments. Your revised manuscript is also likely to be sent to reviewers for further evaluation.

Sincerely,

Alain Kohl

Associate Editor

PLOS Pathogens

Ana Fernandez-Sesma

Section Editor

PLOS Pathogens

Kasturi Haldar

Editor-in-Chief

PLOS Pathogens

orcid.org/0000-0001-5065-158X

Michael Malim

Editor-in-Chief

PLOS Pathogens

orcid.org/0000-0002-7699-2064

While two of the reviewers were positive about the manuscript, the third reviewer still has substantial concerns following revision.

The authors should look at the all comments carefully, and: specifically address the concerns on the comparative analysis in figure 1; clarify the use of Lovo cells and their use; state the numbers of donors as used per experiment and discuss the variability observed; and verify or at least discuss the final comment on validating the lack of infection by qPCR.

Reviewer's Responses to Questions

**Part I - Summary**

Reviewer #1: The authors have addressed all of my concerns and I wish them all the best for publication and their future work.

Reviewer #2: The authors have fully and thoroughly addressed my concerns, as well as the other reviewers', and it is my opinion that the manuscript is sufficient for publication.

Reviewer #3: Whereas the authors described TLR2-mediated activation of monocytes to mature virus in their previous study (PMID 32576819), here they have examined TLR2-mediated activation of monocytes to immature virus. Using pharmacological inhibitors they show that blocking this pathway prevents some of the downstream immune activation. My overall assessment of this revision is that a minor revision was made, when a major revision was needed. Also, the authors did not provide more clarity in the rebuttal document and many questions were answered in a confusing manner when a straightforward answer where to find the improved sections of the text, if any, was needed.

**Part II – Major Issues: Key Experiments Required for Acceptance**

Reviewer #1: N/A

Reviewer #2: (No Response)

Reviewer #3: Major issues remain after revision and some were even introduced with the added data. The addition of infectious virus added in response to Reviewer 3 Question 1 also does not clarify the question about the specificity to prM. Although they have added infectious DENV to Fig. 1 in panels D-E, this is separated from panels B-C and no statistical comparison was made between them that would allow the conclusion that the kinetics are different. Figure 1 legend does not clearly state which serotypes/strains were used and if they are matched for comparison between B-C. No concentrations of virus are provided in the figure legend and it is unclear if the same concentrations of virus were used in B-C versus D-E, so these cannot be directly compared without first establishing the similar inoculation was used and the only difference is prM. (Protein titration and addition of equivalent titers based on that might help). This is a major concern since the author's central conclusion of this manuscript is that immature virus activates inflammatory cascades earlier (on a different time course). I don't think this conclusion has been supported with data yet.

The authors have added LoVo cells to the manuscript methods, but they don’t clearly state if they were used to grow all of the the immature virus in the text. This is very confusing. Since this aspect of the conclusions, that the response is dependent on immature virus is a central conclusion it seems important to do this experiment. If you re-read the response to Reviewer 3 Question 1, this response is not directly answering the question so it's hard to evaluate what the authors have done.

The authors did not address some of the valid reviewer concerns, such as reviewer 1’s request to add more human donors since there was very high variability. That comment was not addressed in the slightest and it was written that 6 donors were used, but this is misleading because there are only 3 in any given experiment.

The authors also don’t validate the lack of infection by negative strand virus PCR, which is a very simple control to do and it’s concerning why they are unwilling to add simple inexpensive controls.

**Part III – Minor Issues: Editorial and Data Presentation Modifications**

Reviewer #1: N/A

Reviewer #2: (No Response)

Reviewer #3: The new Figure 1C is confusing. The authors should show the time course on the x-axis and use the donors as different colors indicated in the figure legend.

I don't think it's accurate to refer to TLR2 as an "axis".

PLOS authors have the option to publish the peer review history of their article (what does this mean?). If published, this will include your full peer review and any attached files.

Reviewer #1: **Yes: **Connor G G Bamford

Reviewer #2: No

Reviewer #3: No
---

## [Editor Report · Decision Letter 2]

4 Oct 2022

Dear Dr. Rodenhuis-Zybert,

We are pleased to inform you that your manuscript 'TLR2 axis on peripheral blood mononuclear cells regulates inflammatory responses to non-infectious immature dengue virus particles' has been provisionally accepted for publication in PLOS Pathogens.

Best regards,

Alain Kohl

Associate Editor

PLOS Pathogens

Ana Fernandez-Sesma

Section Editor

PLOS Pathogens

Kasturi Haldar

Editor-in-Chief

PLOS Pathogens

orcid.org/0000-0001-5065-158X

Michael Malim

Editor-in-Chief

PLOS Pathogens

orcid.org/0000-0002-7699-2064
---

## [Editor Report · Acceptance letter]

10 Oct 2022

Dear Dr. Rodenhuis-Zybert,

We are delighted to inform you that your manuscript, "TLR2 axis on peripheral blood mononuclear cells regulates inflammatory responses to non-infectious immature dengue virus particles," has been formally accepted for publication in PLOS Pathogens.

Best regards,

Kasturi Haldar

Editor-in-Chief

PLOS Pathogens

orcid.org/0000-0001-5065-158X

Michael Malim

Editor-in-Chief

PLOS Pathogens

orcid.org/0000-0002-7699-2064